# TIM3 Checkpoint Inhibition Fails to Prolong Survival in Ovarian Cancer-Bearing Mice

**DOI:** 10.3390/cancers16061147

**Published:** 2024-03-14

**Authors:** Yani Berckmans, Ann Vankerckhoven, Aarushi Audhut Caro, Julie Kempeneers, Jolien Ceusters, Gitte Thirion, Katja Vandenbrande, Ignace Vergote, Damya Laoui, An Coosemans

**Affiliations:** 1Laboratory of Tumor Immunology and Immunotherapy, Department of Oncology, Leuven Cancer Institute, KU Leuven, 3000 Leuven, Belgium; 2Laboratory of Dendritic Cell Biology and Cancer Immunotherapy, VIB Center for Inflammation Research, 1050 Brussels, Belgium; 3Brussels Center for Immunology, Vrije Universiteit Brussel, 1050 Brussels, Belgium; 4Department of Gynaecology and Obstetrics, Leuven Cancer Institute, University Hospitals Leuven, 3000 Leuven, Belgium

**Keywords:** checkpoint inhibitor, ovarian cancer, chemotherapy, preclinical

## Abstract

**Simple Summary:**

Ovarian cancer still ranks as the deadliest gynecological malignancy worldwide. In this research, we aimed to evaluate the potential of anti-TIM3, a relatively novel checkpoint inhibitor, for the treatment of ovarian cancer. Our preclinical studies showed no improvement of survival in ovarian cancer-bearing mice after anti-TIM3 treatment. Additionally, we found no significant immunological changes induced by this therapy in treated mice. However, changing the order of combination treatment with anti-TIM3 and chemotherapy influenced the outcome in mice. Further preclinical studies are required to find and optimize ovarian cancer treatment approaches.

**Abstract:**

Immune checkpoint inhibitor (ICI) therapy has proven revolutionary in the treatment of some cancers. However, ovarian cancer remains unresponsive to current leading ICIs, such as anti-PD1 or anti-PD-L1. In this article, we explored the potential of an upcoming checkpoint molecule, T-cell immunoglobulin and mucin domain 3 (TIM3), for the treatment of ovarian cancer using a syngeneic orthotopic mouse model (ID8-fLuc). Besides therapeutic efficacy, we focused on exploring immune changes in tumor tissue and peritoneal fluid. Our results showed no improvement in survival in ovarian cancer-bearing mice after anti-TIM3 treatment when used as monotherapy nor when combined with anti-PD1 or standard-of-care chemotherapy (carboplatin/paclitaxel). This was reflected in the unaltered immune infiltration in treated mice compared to control mice. Altering the order of drug administration within the combination treatment altered the survival results, but did not result in a survival benefit over chemotherapy alone. These findings highlight the need for further preclinical studies to find beneficial treatment schemes and combination therapies for ovarian cancer.

## 1. Introduction

High-grade serous ovarian cancer (HGSOC), the most prevalent histological subtype of ovarian cancer, remains in the top five most fatal cancers in women worldwide [1,2]. Due to the absence of specific symptoms and effective screening strategies, a majority of the patients are diagnosed in advanced FIGO (International Federation of Gynecology and Obstetrics) stages (FIGO stage III or IV) [2,3]. In addition, more than 70% of HGSOC patients will experience recurrence of the disease after primary treatment consisting of debulking surgery and platin-based chemotherapy [4,5,6]. The 5-year survival rate for advanced HGSOC patients is therefore still only about 30% [7], despite the introduction of two targeted therapies: bevacizumab (anti-vascular endothelial growth factor) [5] and PARP (poly-ADP ribose polymerase) inhibitors [8].

Immunotherapy has revolutionized the world of cancer therapies, with many successes achieved in melanoma and non-small-cell lung cancer (NSCLC) [9]. Multiple clinical trials have shown improved progression-free survival, which resulted in current EMA (European Medicines Agency) approval for seven immune checkpoint inhibitors (ICIs) targeting either CTLA4 (cytotoxic T-lymphocyte associated protein 4), PD1 (programmed cell death protein 1) or PD-L1 (programmed death-ligand 1) [9,10,11].

However, in ovarian cancer patients, ICI therapy has not proven successful so far. In monotherapy, multiple ICI therapies have been tested in phase I/II trials, including atezolizumab (NCT01375842), pembrolizumab (KEYNOTE-100, NCT02674061), and avelumab (NCT01772004), all showing disappointing results [12,13,14]. Consequently, combination treatment of ICI with chemotherapy has been explored thoroughly for the treatment of ovarian cancer. Despite all efforts, four phase III studies were recently published showing no benefit [15,16,17,18]. More recently, disappointing interim results of the ANGOT-Ov41/ANITA phase III trial (NCT03598270) were presented at the ESMO Congress 2023, in which atezolizumab was combined with chemotherapy and PARP inhibition [19].

Clinical trials in ovarian cancer have mainly focused on anti-PD1/PD-L1 therapy. However, we and others have demonstrated that the immune biology of ovarian cancer is more complex and that innate immunosuppression may be a major barrier to successful treatments [20,21,22,23]. The current clinically exploited checkpoint inhibitor therapies do not focus on this. In this paper, we explore the potential of targeting the checkpoint receptor TIM3 (T-cell immunoglobulin and mucin-domain containing 3) as a treatment for ovarian cancer. TIM3 or CD366 is a type 1 transmembrane protein that was first associated with T-cell exhaustion due to its discovery on CD4+ and CD8+ T cells [24]. An advantage of TIM3 inhibition is the selective expression of TIM3 on intratumoral T cells, reducing the unwanted effects on T-cell responses outside of the tumor that are often reported after anti-PD1 or anti-CTLA4 treatment [25]. Additionally, unlike some other immune checkpoints that are mainly expressed on adaptive immune cells, TIM3 is found to be constitutively expressed on a variety of innate immune cells, including macrophages, monocytes, natural killer cells, and dendritic cells (DCs) [26]. Therapeutic interventions targeting this checkpoint molecule can therefore have a large impact on the innate immune cells in the tumor microenvironment (TME).

Multiple ligands have been identified to interact with the TIM3 checkpoint receptor, including galactin 9, carcinoembryonic antigen-related cell adhesion molecule 1 (CEACAM-1), phosphatidylserine, and high-mobility group protein B1 (HMGB1) [24,25]. HMGB1 was shown to be critical in the TIM3-mediated inhibition of tumor-associated DCs after nucleic acid release from dying cancer cells [27]. CEACAM-1 binding with TIM3 was shown to regulate TIM3-associated autoimmunity and anti-tumor immune response [24]. Interaction of galactin 9 was shown to induce apoptosis of TIM3-expressing Th1 T cells, as well as promote accumulation of myeloid-derived suppressor cells (MDSCs) in the TME [25]. So far, the anti-TIM3 ICI has demonstrated success in multiple preclinical models for colon carcinoma, prostate cancer, and acute myelogenous leukemia (AML) [25,28]. Many TIM3-targeting antibodies have been developed and are being investigated in first-in-human clinical trials in both solid tumors as well as AML and myelodysplastic syndrome (MDS). Clinical combination with anti-PD1 is often included in these trials, in line with the preclinical indication. A TIM3 ICI, TSR-022 showed promising results when combined with anti-PD1 in NSCLC patients, showing progression after previous anti-PD1 treatment (NCT02817633) [28]. Preliminary results in high-risk AML or MDS patients showed beneficial outcomes after combined treatment with anti-TIM3 (MBG453) ICI and a hypomethylating therapy (NCT03066648) [29].

In ovarian cancer, no clinical studies have been conducted testing the efficacy of TIM3 immune checkpoint inhibition so far. One phase I/IIb trial (NCT02608268) showed a tolerable profile of anti-TIM3 treatment alone and in combination with an anti-PD1 ICI in patients with advanced solid tumors, of which 17% were of ovarian origin, although no follow-up study has been reported so far [30]. However, increased TIM3 expression on immune cells has been associated with a more advanced disease stage and decreased survival in ovarian cancer patients [24,31]. This prognostic value of TIM3 could be associated with a suppressed immune response in HGSOC, suggesting that TIM3 expression may be important as a biomarker in this disease [32]. Based on a literature search, a total of 45 publications (both published papers and conference abstracts) discussed immune checkpoint expression levels in ovarian cancer. Most data concerned immune infiltrate analysis on tumor biopsies of ovarian cancer patients. Despite large heterogeneity between study types, an average of 20% positivity in marker expression was reported for TIM3, which is in contrast with the more highly expressed PD1 levels (on average by 50% of the investigated cells) (Figure 1, additional information in Appendix A). However, in a study by Blanc-Durand et al., a significantly higher expression of TIM3 (76%) was observed compared to PD-L1 (28%) on ovarian cancer tumor samples of 90 and 173 patients, respectively. Interestingly, they observed no change in TIM3 levels induced by platinum-based chemotherapy, which did increase the percentage of PD-L1-positive tumors [33].

Additionally, multiple studies have indicated the co-expression of TIM3 and PD1 on exhausted T cells in ovarian cancer, specifically in the tumor and ascites [24]. Inhibition of the checkpoint molecule TIM3 can therefore be a prospective therapeutic target, possibly even in combination with co-inhibition of the PD1 checkpoint molecule. In this research article, we further explore the potential of anti-TIM3 ICI in ovarian cancer-bearing mice.

## 2. Materials and Methods

### 2.1. Cell Culture

In-house ID8-fluc cells were cultured in Dulbecco’s modified Eagle’s medium (Gibco, Thermo Fisher Scientific, Waltham, MA, USA) supplemented with 10% fetal calf serum (FCS), 100 U/L penicillin–streptomycin, 2 mM L-glutamine, 2.5 µg/mL amphotericin B and 10 mg/mL gemcitabine. All cultures were maintained at 37 °C in the presence of 5% CO_2_. When 80–100% confluency was reached, cells were harvested using 0.05% trypsin–EDTA (Gibco) and suspended in Dulbecco’s phosphate-buffered saline (DPBS, Gibco) for inoculation.

### 2.2. Mouse Model and Treatment

Female six- to eight-week-old C57Bl/6 mice obtained from Envigo (Horst, The Netherlands) were inoculated intraperitoneally with 5 × 10^6^ ID8-fluc cells, leading to the development of a stage III–IV ovarian cancer model. Murine experiments were approved by the KU Leuven ethics committee (P125/2017). Guidelines for ethical standards were followed according to the NIH *Guidelines for the Care and Use of Laboratory Animals*, EU Directive 2010/63/EU as amended by Regulation (EU) 2019/1010, and the Animal Research: Reporting of In Vivo Experiments (ARRIVE) guidelines. Relief of distress due to ascites accumulation, a symptom of advanced ovarian cancer, was performed by drainage of the ascites fluid when mice reached 32 g of weight. First ascites drainage was used as a measure to evaluate symptom development and disease progression. Additionally, all mice were evaluated for overall survival based on the previously published humane endpoint criteria by Baert et al. [34].

Treatment schedules are displayed in Figure 2. Chemotherapy consisted of carboplatin (100 mg/kg) combined with paclitaxel (10 mg/kg) and was administered intraperitoneally on day 21 post-inoculation. Immune checkpoint inhibitor treatments were administered through intraperitoneal injection. The anti-PD1 checkpoint inhibitor (BioXcell (clone RMP1-14)) was administered (50 µg/injection) every two days for a total of five administrations starting on day 20 post-inoculation. Anti-TIM3 was obtained via GSK (formerly TESARO) (clone SR13167) and BioXcell (clone RMT3-23) and was administered (350 µg/injection) following a biweekly schedule for three to four weeks, as indicated in Figure 2. The anti-TIM3 dose was selected based on a dosing study performed previously in this model. All treatments were diluted using DPBS before injection. Control mice received DPBS vehicle administration. Two administration schedules were tested, starting either on day 20 (in the simultaneous scheme) or day 28 (in the sequential scheme). A total of five in vivo experiments were performed, in which different treatment combinations and administration schedules (shown in Figure 2) were tested.

### 2.3. Immune Monitoring

To determine the immune composition, mice were euthanized at predefined time points in one experiment (day 35 and day 49 post-inoculation). A peritoneal washing was performed using 10 mL of DPBS (Gibco) to collect the immune cells present in the peritoneal TME for flow cytometric analysis, a technique that was described earlier by our group [22]. Additionally, peritoneal biopsies were collected in 4% paraformaldehyde and embedded in paraffin for subsequent immunohistochemical staining.

Flow cytometry analysis was performed on immune cells isolated from peritoneal washings. Single cell suspension was prepared by treatment with Ammonium-Chloride-Potassium erythrocyte lysis buffer, in case macroscopic blood was present. Single cell suspensions were resuspended in Hanks’ Balanced salt solution (HBSS) and incubated with Fixable Viability Stain 575V (1:2000, BD Biosciences, Franklin Lakes, NJ, USA, cat# 565694) for 15 min at room temperature in the dark. Next, cell suspensions were washed with HBSS and resuspended in HBSS with 2 mmol/L EDTA and 0.5% (*v*/*v*) FCS. To prevent nonspecific antibody binding to Fcγ receptors, cells were preincubated with CD16/CD32-specific antibody (clone 2.4G2, BD Biosciences, cat# 553142). Cell suspensions were then incubated with fluorescently labeled antibodies (Appendix A) diluted in HBSS with 2 mmol/L EDTA and 0.5% (*v*/*v*) FCS for 20 min at 4 °C and then washed with the same buffer. For intracellular staining (all antibodies except IFNγ), samples were centrifuged at 450 g and fixed using theIntracellular Fixation & Permeabilization Buffer Set (Thermo Fisher Scientific, Waltham, MA, USA, 88–8824–00) according to the manufacturer’s instructions. For intracellular staining of IFNγ, single-cell suspensions were first incubated in an ex vivo culture medium (RPMI containing 10% (*v*/*v*) FCS, 300 μg/mL l-glutamine, 100 units/mL penicillin, 100 μg/mL streptomycin, 1% (*v*/*v*) MEM nonessential amino acids, 1 mmol/L sodium pyruvate, and 0.02 mmol/L 2-mercaptoethanol) containing Golgiplug (1:1000, BD Biosciences, cat# 555029) for four hours at 37 °C before fixation and permeabilization as described above, and staining of intracellular IFNγ. Flow cytometry data were acquired using a BD FACSymphony™ A3 (BD Biosciences) and analyzed using FlowJo (v10.10.0). The gating strategy to identify immune cell populations in tumors is shown in Appendix A.

Immunohistochemical staining was performed on collected peritoneal biopsies. Tumor slices were first deparaffinized and rehydrated by heating at 60 °C for 60 min and subsequent washing with xylene, followed by a series of graded ethanol until distilled water. Next, the tissue samples were treated with an epitope retrieval solution (pH: 9) at 97 °C in warm water bath for 30 min and washed in tris-buffered saline (TBS). Unspecific binding sites were blocked using a BloxallTM blocking solution (VectorLabs, Newark, NJ, USA) and Normal Horse Serum (2.5%, VectorLabs). The first staining was performed using the primary antibody, rabbit anti-CD4 (Abcam, Cambridge, UK, ab183685 1:150), at room temperature for one hour before washing with TBS 0.04% Tween and incubating for 30 min with the secondary antibody (anti-rabbit-HRP, VectorLabs). 3-Amino-9-Ethylcarbazole (AEC) substrate was incubated for 10 to 60 min at room temperature until chromogen development. The second sequential staining was performed by stripping the AEC substrate using graded ethanol washing and subsequent antibody elution at 56 °C for 30 min with a 2-Mercaptoethanol, 10% (*w*/*v*) SDS, 0.5 M tris-HCl buffer. Afterwards epitope retrieval and blocking was performed as described earlier. Staining was completed using rabbit anti-CD20 (Abcam, ab64088 1:100) with an anti-rabbit-HRP secondary antibody and AEC substrate. Lastly, similar sequential staining was performed after AEC substrate stripping, epitope retrieval and blocking, using rabbit anti-CD8 (Abcam, ab237723 1:250) primary antibody before application of the ImmPRESS Excel Amplified Polymer Staining kit (MP-7601, VectorLabs) as described by the manufacturer. All immunohistochemically stained slides were counterstained using hematoxylin and scanned using the Aperio Versa slide scanner (Leica Biosystems, Nussloch, Germany). For the evaluation of the immune cell density, two regions of interest (ROI) were selected for each tumor sample using QuPath 0.4.4 [35]. Tumoral areas were visually discriminated against non-tumoral areas. All ROI were copied for each sequential immunohistochemical staining (CD20, CD4 and CD8) of the same tissue slide. Positive cells were identified and counted in each ROI. Absolute positive cell numbers were used to calculate the mean positive cell density per mm^2^ tumor area.

### 2.4. Statistics

A statistical power analysis was performed to determine the sample size for the primary endpoint and immune monitoring purposes to reach a power of at least 0.80. Survival curves were compared using the log-rank test. Adjustment for multiple comparisons was performed with the Benjamini-Hochberg procedure. Flow cytometry data was summarized with means and standard deviations and visualized using bar charts. For comparison between the different treatment groups, an ANOVA with Tukey’s multiple comparisons test was used. Immunohistochemical data was visualized with boxplots. In the analyses, an (adjusted) *p* value < 0.05 was considered significant. All analyses were performed using GraphPad Prism version 10.1.0 and R version 4.1.0.

## 3. Results

### 3.1. Anti-TIM3 Is Unable to Prolong Survival as Monotherapy as Well as in Combination with Anti-PD1 in Ovarian Cancer-Bearing Mice

First, anti-TIM3 monotherapy was compared to anti-PD1 monotherapy in tumor bearing mice. Both ICI treatments were administered intraperitoneally with injections starting on day 20 after tumor inoculation. We observed no difference in survival and ascites accumulation between the treated groups nor when compared to the control group (Figure 3A) (Appendix A). Additionally, no toxicity was visible following anti-TIM3 administration, as shown in the weight curves (Appendix A).

Next, the two checkpoint inhibitors were combined, since literature suggests a potential synergy due to the concurrent expression of both markers on exhausted T cells. However, in ID8-fLuc bearing mice, this combination treatment significantly reduced survival compared to control mice (*p*-value: 0.0064) (Figure 3B). This decrease was not significant when analyzing ascites accumulation between both groups (Appendix A). Therefore, we focused on further exploring the potential of the anti-TIM3 without the addition of anti-PD1.

### 3.2. Simultaneous Administration of Standard-of-Care Chemotherapy and Anti-TIM3 Checkpoint Inhibition Does Not Result in a Synergistic Effect in Ovarian Cancer-Bearing Mice

We next combined the standard of care chemotherapy consisting of carboplatin and paclitaxel with anti-TIM3 checkpoint inhibition. In this experiment, both treatments were started simultaneously as shown in Figure 2 (simultaneous scheme; day 20/24/27/31/34/38). This is the treatment schedule most often explored for combined immune-oncological therapy [36]. Figure 4 shows a significant increase in survival in both treatment groups compared to vehicle control treated mice. However, this effect most likely relates to the chemotherapy treatment, as the statistical analysis of both Kaplan-Meier curves showed no significant difference between the treatment groups (*p*-value: 0.173). Moreover, the median survival of mice receiving anti-TIM3 in combination was 110 days compared to chemotherapy only treated mice (98.5 days). Similarly, no significant difference in ascites accumulation was noted between both chemotherapy only and combination treated mice (Appendix A).

### 3.3. Order of Anti-TIM3 and Chemotherapy Administration Influences Survival of Ovarian Cancer-Bearing Mice without Overall Improvement over Chemotherapy Alone

A literature review by our group demonstrated that the order of the treatments can influence the survival outcome [36]. Therefore, different administration orders of the combination with carboplatin-paclitaxel and TIM3 inhibition were explored in the following experiment. Similar to the previous set-up, chemotherapy was administered in a single injection on day 21. However, biweekly anti-TIM3 administration was given either simultaneously starting on day 20 or sequentially starting on day 28, following the administration schemes shown in Figure 2. Survival is displayed in Figure 5.

In this experiment, the survival curve of the simultaneous combination group did not differ from mice treated with chemotherapy alone (*p*-value: 0.9645). Moreover, the sequential combination treatment appeared to abrogate the positive survival benefit caused by chemotherapy alone, although this was not significant (*p*-value: 0.0595). A significant prolongation of the survival was observed between the control group and both the chemotherapy only treated mice (*p*-value: 0.0025) and the simultaneous combination group (*p*-value: 0.0025). Additionally, statistical significance was also noted when comparing the survival of anti-TIM3 only treated mice compared to both treatment groups (*p*-value: 0.001 and 0.0015, respectively). These beneficial survival outcomes can likely be attributed to the chemotherapy effect, as addition of anti-TIM3, both in simultaneous or sequential administration schemes, did not result in further improvement. Similar observations could be made when displaying the first ascites drainage at 32 g as a measure of survival (Appendix A). Here, no significant difference was noted between both combination treated groups and chemotherapy alone. However, the ascites accumulation appeared to be slower in the simultaneous treated mice compared to mice receiving only chemotherapy, although this was not statistically significant. This effect of chemotherapy treatment could also be observed in the weight curves in which an increase (ascites, tumor progression) appeared at a later time in all chemotherapy treated groups compared to the controls and the anti-TIM3 monotherapy group. Moreover, compared to the sequential group, this increase was observed even later in de chemotherapy only and simultaneous group (Appendix A). Interestingly, this experiment showed a trend towards a difference between simultaneously treated and sequentially treated mice when analyzing both survival and ascites accumulation which, although not significant, may indicate to the importance of treatment order.

### 3.4. Changing the Combination Treatment Schedule Does Not Significantly Alter Immune Cell Composition in Peritoneal Washings of Tumor Bearing Mice

Next, we repeated the comparison of both simultaneous and sequential treatment schemes to explore a potential immunological explanation for the difference in outcome observed between both treatment groups (Figure 5, Appendix A). Mice were euthanized at predefined time points (day 35 and day 49) to perform immune monitoring. Differences in immune cell composition were then analyzed using high-dimensional flow cytometry on peritoneal washings and immunohistochemical analysis on peritoneal biopsies following the gating strategies depicted in Appendix A.

The analysis of the immune cell subsets is displayed in Figure 6. No significant change was observed in the different subsets of the T-cell compartment within the total CD45+ cell population. Statistical differences for CD4+ and CD8+ T cells within the total T-cell population were observed in all treatment groups compared to control mice. Similarly, differences in regulatory T cells (Tregs) within the total T-cell population were noted in the two combinatorial treatment groups, but not within the total CD45+ cell population. The percentage of Tregs within the total T-cell population appeared to be increased in the sequential combination-treated mice compared to controls on day 49. In the simultaneous combination group, a decrease of Tregs within the total T-cell population was seen between analyses at day 35 and day 49. B cells within the total CD45+ population were constant over time and between different treatment groups.

Natural killer cells (NK1.1+) were decreased in all treated groups compared to controls on day 35. Moreover, in the control group, NK1.1+ cells were significantly lower at day 49 compared to day 35. The conventional dendritic cell 2 (cDC2) population showed a decreased level in chemotherapy-treated mice compared to controls on day 35.

Overall, significant changes appeared to be correlated with chemotherapy treatment rather than checkpoint blockade, as addition of anti-TIM3 treatment to chemotherapy resulted in no added difference.

The level of PD1, PD-L1 or TIM3 expression was investigated on different immune cell subsets and is shown as the difference in mean fluorescence intensity (dMFI) between population and FMO (fluorescence minus one) (Appendix A). Both combination groups presented a lower (insignificant) dMFI of PD1 on Tregs on day 35. A significant change in dMFI of PD1 expression on Tregs was observed in chemotherapy only-treated mice compared to control mice. Due to a limitation in available cells, no analysis of PD1 expression on Tregs could be performed on day 49. The dMFI value of TIM3 expression on B cells was increased in all treated cells compared to controls, which was only analyzed on day 49. Levels of dMFI of the checkpoint marker PD-L1 were affected on cDC1, cDC2 and monocytes. However, this effect seemed to be dependent on disease progression (time) rather than the administered treatment.

The presence of activation and exhaustion markers on CD4+ and CD8+ T cells can be seen in Appendix A. In line with previous observations, significant changes could generally be correlated with disease progression or the administration of chemotherapy.

### 3.5. Immunohistochemical Analysis Shows No Change in Amount or Spatial Distribution of T and B Cells after Combined Treatment

Immunohistochemical analysis was performed on peritoneal biopsies of mice at predefined time points (day 35 and day 49 post-inoculation). Tissue samples were stained for CD8, CD4 and CD20 markers. The total number of positively stained cells was measured per tumor area. Representative images of all staining procedures are displayed in Figure 7 and Appendix A.

The presence of CD8+ T cells in peritoneal tumor biopsies was similar in the different treatment groups to the control mice (Figure 8). An increase in intratumoral CD4+ T cells in mice treated with the sequential combination on day 35 was observed. Additionally, CD4+ T cells increased slightly in simultaneously treated mice on day 49 compared to controls. Lastly, higher numbers of intratumoral CD20+ B cells were counted in tumor lesions of sequential combination-treated mice on day 49 after inoculation compared to mice treated with the simultaneous regimen. Unfortunately, due to the small sample size and large variation, no statistical significance could be reached (Figure 8). 

Overall, T cells and B cells were only found intratumorally. No immune cells were present in the surrounding stromal tissue. Additionally, no suspected tertiary lymphoid structure (TLS) aggregates were found in these tissue samples. The overall spatial distribution was the same in all treatment groups and therefore remained unaffected by the different therapies.

## 4. Discussion

Clinical trials testing immune checkpoint inhibition in ovarian cancer have not produced the expected positive results so far. However, the aforementioned studies mainly focused on anti-PD1 or anti-PD-L1, while the checkpoint molecule TIM3 was found to be more widely expressed on a variety of immune cells. Additionally, TIM3 expression has been correlated with reduced survival in ovarian cancer patients [24]. In this article, we therefore aimed to explore the potential of anti-TIM3 checkpoint inhibition for the treatment of ovarian cancer.

We first evaluated the efficacy of anti-TIM3 monotherapy in ovarian cancer-bearing mice. Similar to anti-PD1 monotherapy, no beneficial effect was observed compared to controls. Dual blockade of both TIM3 and PD1 checkpoint molecules resulted in decreased survival compared to controls, despite literature suggesting possible synergy [24] and this being a strategy often chosen in clinical trials [37]. Nevertheless, the mice receiving dual-checkpoint blockage in our experiment showed similar symptom development, characterized as ascites accumulation up to 32 g, to controls (Appendix A). However, after first ascites drainage, mice receiving the ICI combination showed a faster progression towards cachexia than control mice, which resulted in significant differences in overall survival. This rapid decline and decreased survival rate may be related to toxicity of the combined ICI. Toxicity as a result of dual-checkpoint blockade has previously been reported by Martins et al. in patients receiving combined anti-CTLA4 and anti-PD1. In this study, increased endocrinopathy, nephritis, liver toxicity and pneumonitis were associated with the dual-ICI therapy [38]. These results are contrary to the beneficial effects with this dual-ICI combination reported by Sakuishi et al. in a preclinical study using a subcutaneous mouse model for colon carcinoma [39]. Of note, this was another cancer type, and our studies were based on findings in an orthotopic mouse model of advanced ovarian cancer.

To further explore the potential of anti-TIM3, we hypothesized that induction of an immune response could be required. Hence, we focused the following experiments on the combination with standard-of-care chemotherapy for ovarian cancer consisting of carboplatin and paclitaxel. Our team has previously demonstrated that this chemotherapy, both in patients and in mice, reduces immunosuppression in ovarian cancer [21,40]. Moreover, we recently also highlighted the importance of the treatment order when designing immune-oncological combination therapy [36]. Therefore, both a simultaneous therapy regimen and a sequential one were evaluated. However, both combination schedules were unable to significantly increase survival or delay ascites accumulation in mice compared to chemotherapy alone. Overall, the current treatment combination and administration schedules tested in this study did not show a potential of anti-TIM3 therapy for ovarian cancer. This was confirmed by our immunological analysis, where only very minor differences were depicted.

In contrast to our results, Guo et al. studied the treatment of anti-TIM3 in a similar intraperitoneal mouse model for ovarian cancer (ID8), where they observed a significant prolongation of survival when injecting anti-TIM3 in monotherapy on day three after tumor inoculation. However, when the ICI was administered on day 10, no beneficial outcome was reached [41]. This can be an argument for increased potential of this ICI treatment in earlier disease stages. However, we evaluated the efficacy of the treatment only in later disease stages in our experiments, as our targeted group of HGSOC patients are mostly diagnosed at advanced stages [2]. In that study, very hopeful results were accomplished with the combination of anti-TIM3 and CD137 activation, which was not investigated in our study. Interestingly, through depletion experiments, CD8+ T cells were identified to play a crucial role in the anti-tumor response elicited by this treatment [41]. Our results show a decrease in CD8+ T cells in all chemotherapy-treated groups (also in combination with anti-TIM3), which may provide an explanation for the absent survival effect of combining TIM3 inhibition with chemotherapy.

Our team has previously shown the importance of innate immunosuppression, specifically through the presence of MDSC, in ovarian cancer progression [22,42]. Blocking only one checkpoint receptor may therefore be insufficient to overcome this high level of immunosuppression seen in the ovarian cancer tumor microenvironment. Additionally, Tao et al. reported the ability of MDSC to produce galactin 9, a TIM3 ligand, through which CD8+ T-cell exhaustion may be stimulated [43]. However, current murine anti-TIM3 ICI, including the clone used in our studies (RMT3-23) often blocks other ligand interaction sites, such as HMGB1, but not galectin 9 [44,45]. This could result in continued immunosuppression and may explain the poor outcomes in our survival experiments. It could therefore be interesting to design novel TIM3-inhibiting antibodies targeting a broader function of this checkpoint molecule before more clinical trials are initiated.

Nevertheless, also in our experiments, differences in survival following different orders of combination (sequential vs simultaneous) were apparent and indicate an important impact of the administration schedule on therapeutic success rates. We observed decreased survival in mice treated with a sequential combination of chemotherapy and anti-TIM3 compared to chemotherapy only-treated mice and mice receiving the simultaneous combination regimen. Addition of sequential anti-TIM3 therefore seemed to abrogate the positive effect of chemotherapy. This decreased outcome compared to chemotherapy only-treated mice was less apparent when evaluating ascites accumulation. However, in this analysis, the difference between both combination treatment regimens (simultaneous vs sequential) was prominent as well. Although no immunological explanation was found for the decreased outcome in mice treated with the sequential combination of chemotherapy and anti-TIM3, this result is similar to what was observed in patients. The survival outcome in the JAVELIN ovarian 100 trial (NCT02718417) showed a trend towards a decreased survival in patients receiving chemotherapy (carboplatin–paclitaxel) followed by sequential administration of avelumab (anti-PD-L1) compared to patients receiving only chemotherapy [16].

We acknowledge some limitations in our study, such as the group sizes used in survival experiments. Due to treatment-induced toxicity, preliminary euthanasia of mice reduced this number. Another limitation is the limited subset of immune cells (CD8+, CD4+ and CD20+ cells) that were analyzed using immunohistochemical staining of the tumor tissue. Differences between other immune cell populations may have been missed, although they have been extensively explored in peritoneal washings using flow cytometry. As in all preclinical research, it is imperative to acknowledge the translational gap when using mouse models to study therapeutic response. Moreover, studies have indicated a lower immune infiltration being present in ID8 bearing mouse compared to other ovarian cancer mouse models [46]. In addition, the ID8-fLuc-derived mouse model lacks p53 and BRCA1/2 mutations, which have also been shown to produce distinct properties and impact the TME. However, we have demonstrated a similar abundant immunosuppression in the ID8-fLuc model to that in patients [22], as well as a similar response to first-line chemotherapy and anti-PD1 checkpoint inhibition to that seen in clinical trial results so far [47]. Nevertheless, it may be relevant to perform additional testing in models with phenotypic differences to the ID8-fLuc model to simulate the heterogeneity in human cancers and the model independence of preclinical experimental findings.

We realize that some questions might remain unanswered; however, as we were unable to induce a survival benefit with anti-TIM3 in the different tested combination settings, it seemed unethical, in the current changing time of refinement, replacement, and reduction of preclinical in vivo research [48], to continue these experiments. Underscoring our observations is the fact that although multiple clinical phase I/II trails have been initiated to investigate TIM3 blockade in solid tumors from 2015 onwards [28], no study has progressed to phase III.

## 5. Conclusions

In conclusion, our results indicate that anti-TIM3 checkpoint inhibition may be insufficient to induce a beneficial outcome as monotherapy for ovarian cancer. Furthermore, the combination schedules with either anti-PD1 or standard-of-care chemotherapy tested in the ID8-fLuc mouse model does not provide the envisaged response. Changing the order of immune-oncological combination treatment can impact the outcome, but did not solve the lack of anti-TIM3 efficacy for ovarian cancer treatment in this study. Additional preclinical experiments towards more promising combination regimens could reveal the potential of this type of immunotherapy as a clinical treatment for ovarian cancer patients.

## Figures and Tables

**Figure 1 cancers-16-01147-f001:**
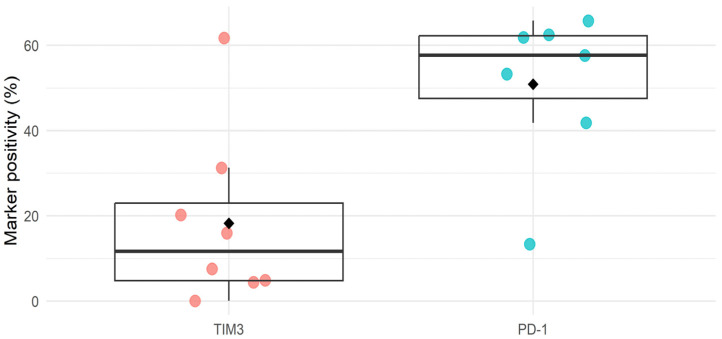
Summary figure of literature study on expression of TIM3 and PD1 in ovarian cancer tissue. Dots are individual study values, diamonds represent the mean. Additional information can be found in Appendix A.

**Figure 2 cancers-16-01147-f002:**
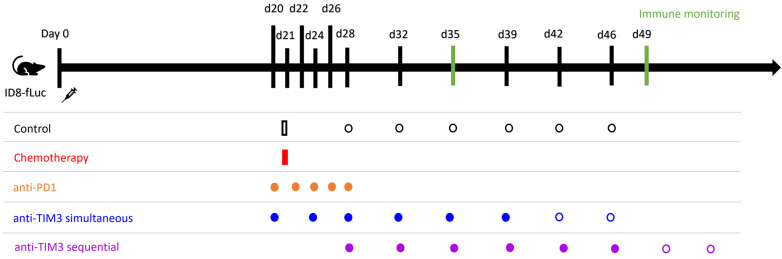
Administration schedule of immune checkpoint inhibitors in monotherapy and of the different combination treatments tested in vivo in ID8-fLuc bearing mice. Intraperitoneal injection was used for all administrations. All control mice received vehicle control injections with DPBS. Chemotherapy consisted of paclitaxel (10 mg/kg) and carboplatin (100 mg/kg) and was administered once at day 21 post-inoculation. Anti-PD1 (50 µg/injection) was administered on days 20/22/24/26/28 after tumor inoculation. Anti-TIM3 (350 µg/injection) was tested in different administration schedules over various experiments performed. Administration in a simultaneous scheme was defined as injections on day 20/24/27/31/34/38 +/− day 41/45 (three to four weeks of treatment). The sequential treatment scheme for TIM3 similarly followed a biweekly schedule starting on day 28 for three to four weeks (day 28/32/35/39/42/46 +/− 49/53).

**Figure 3 cancers-16-01147-f003:**
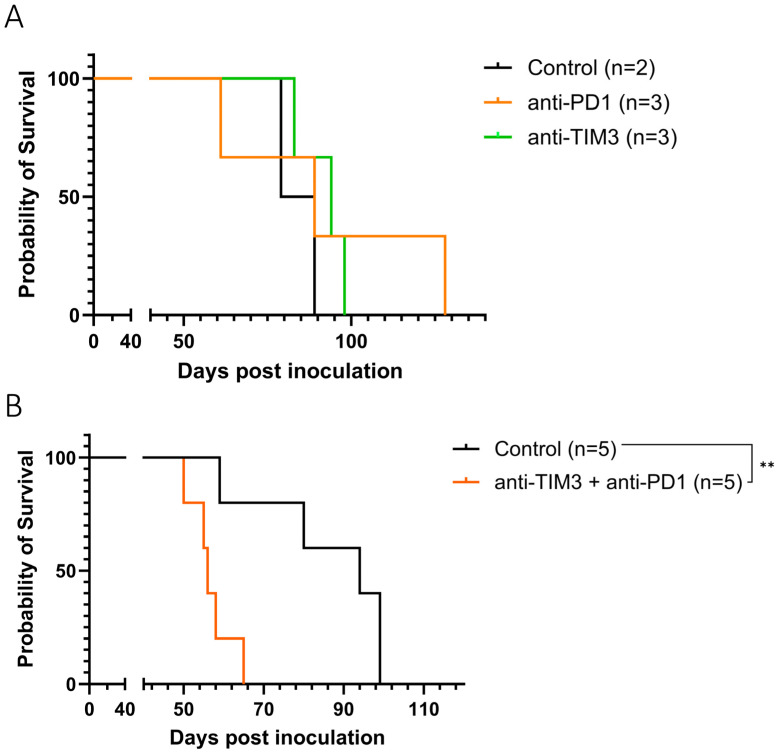
Kaplan-Meier curve showing the survival of ID8-fLuc bearing mice following monotherapy (**A**) or combined immune checkpoint inhibition (**B**). Mice were treated with either vehicle control injections with DPBS, anti-PD1 (50 µg, day 20/22/24/26/28) or anti-TIM3 (350 µg, sequential treatment scheme, day 20/24/27/31/34/38) or a combination of both immune checkpoint inhibitor treatments, intraperitoneally. ** *p*-value: 0.0064.

**Figure 4 cancers-16-01147-f004:**
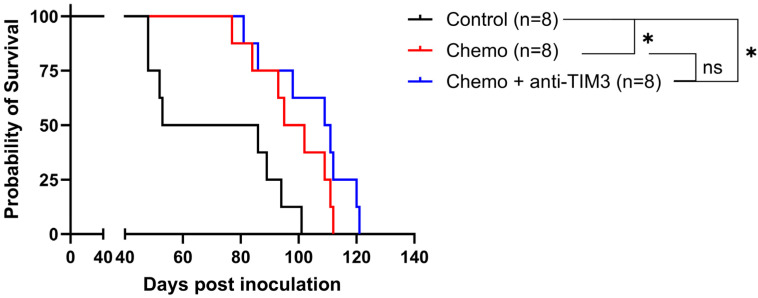
Kaplan-Meier curve showing the survival of ID8-fLuc bearing mice following simultaneous administration of chemotherapy and anti-TIM3. All mice were treated with either vehicle control, chemotherapy consisting of carboplatin (100 mg/m)g and paclitaxel (10 mg/kg) or a combination of chemotherapy and anti-TIM3 ICI in the simultaneous treatment scheme (350 µg, day 20/24/27/31/34/38), through intraperitoneal injections. * *p*-value < 0.05, ns = not significant.

**Figure 5 cancers-16-01147-f005:**
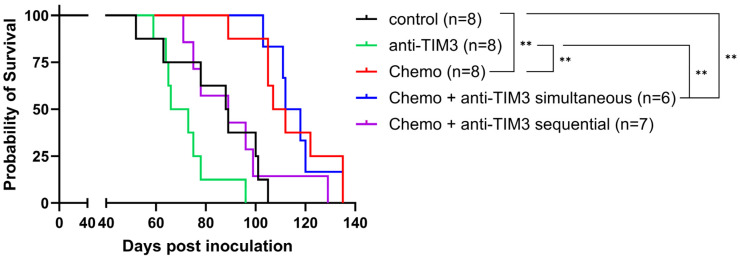
Kaplan-Meier survival curve on ID8-fLuc bearing mice following different combination administration schemes with chemotherapy and anti-TIM3 treatment. Mice were treated with either vehicle control, anti-TIM3 (350 µg/injection) in monotherapy following a biweekly administration for four weeks (D28/32/35/39/42/46/49/53) or in combination with chemotherapy consisting of carboplatin (100 mg/kg) + paclitaxel (10 mg/kg) on day 21 with anti-TIM3 administered in either the sequential treatment schedule (D28/32/39/42/46/49/53) or a simultaneous administration scheme (D20/24/27/31/34/38/41/45). ** *p*-value < 0.005.

**Figure 6 cancers-16-01147-f006:**
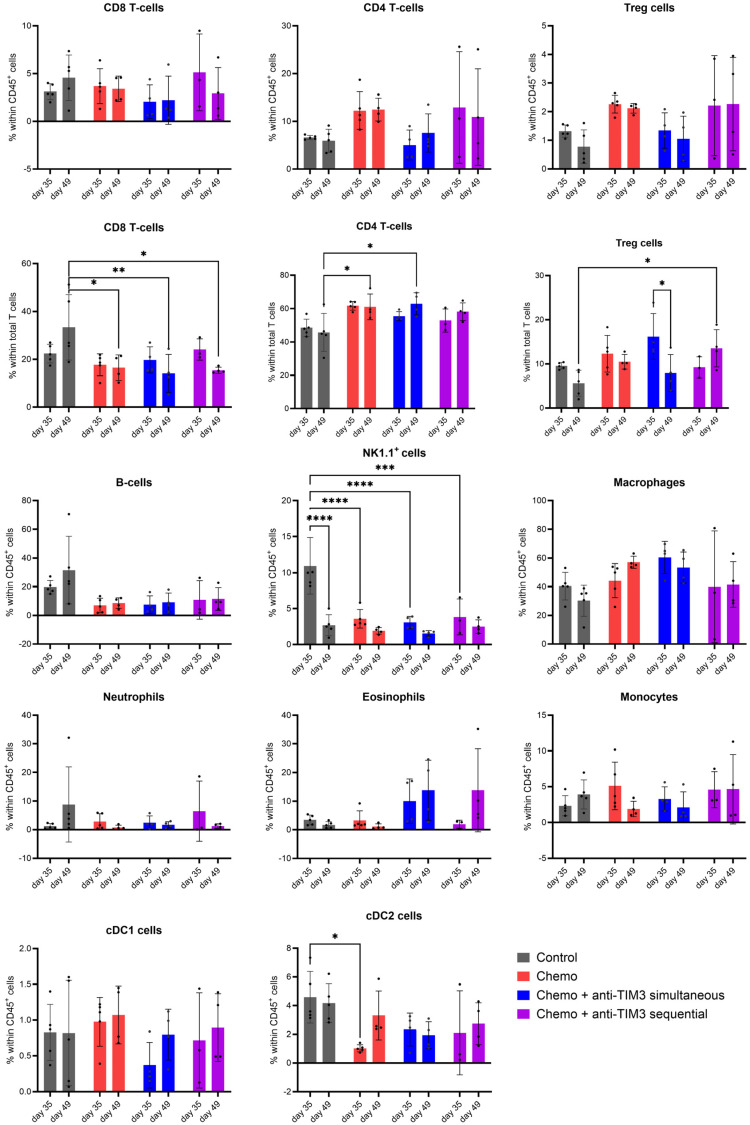
Bar charts showing the immune cell composition in peritoneal washings of ID8-Fluc-bearing mice, analyzed through flow cytometry. Mice received either vehicle control, chemotherapy consisting of carboplatin (100 mg/kg) + paclitaxel (10 mg/kg) injected on day 21 post-inoculation or a combination of chemotherapy and anti-TIM3 following either a simultaneous (D20/24/27/31/34/38) or a sequential administration schedule (D28/32/35/39/42/46) for three weeks. Peritoneal washings were performed postmortem at predefined time points post-inoculation (day 35 and day 49). Treg: regulatory T cells, cDC1/2: conventional dendritic cell 1/2, NK1.1+ cells: natural killer cells 1.1. * *p*-value < 0.05, ** *p*-value < 0.01,*** *p*-value < 0.001, **** *p*-value < 0.0001.

**Figure 7 cancers-16-01147-f007:**
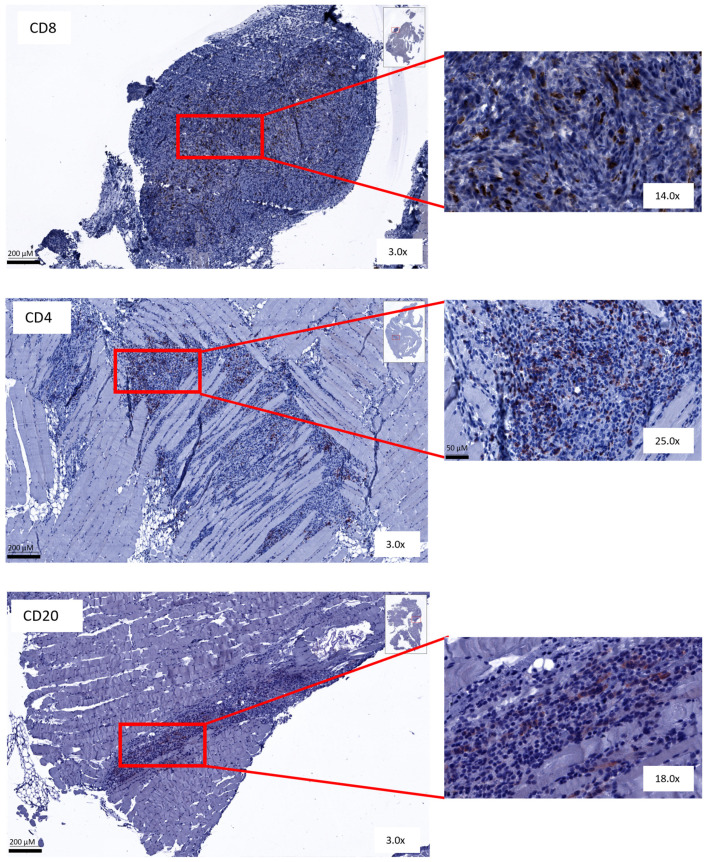
Representative figures of immunohistochemical staining. Peritoneal biopsies of mice stained for CD8+, CD4+ and CD20+ cells.

**Figure 8 cancers-16-01147-f008:**
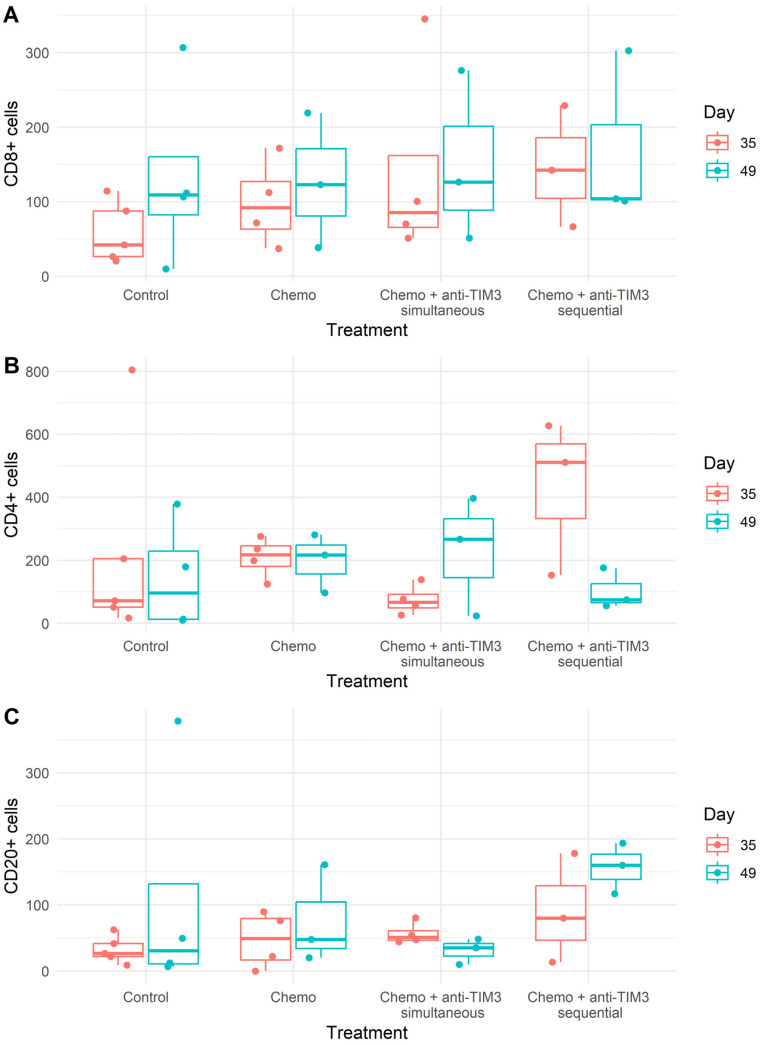
Box plots showing the average number of positively stained ((**A**) CD8^+^; (**B**) CD4^+^; (**C**) CD20^+^) cells per mm^2^ of tumor area in peritoneal biopsies. Mice were treated with carboplatin (100 mg/kg) + paclitaxel (10 mg/kg) chemotherapy with or without anti-TIM3 (350 µg/injection) in different administration schedules. Simultaneous administration (D20/24/27/31/34/38) or sequential administration (D28/32/35/39/42/46).

## Data Availability

The original contributions presented in the study are included in the article/Appendix A, further inquiries can be directed to the corresponding author/s.

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
