# Peer review of "TIM3 Checkpoint Inhibition Fails to Prolong Survival in Ovarian Cancer-Bearing Mice"

_cancers, 2024, doi:10.3390/cancers16061147_

Round 1

Reviewer 1 Report

Comments and Suggestions for Authors

In this article, mice with ovarian cancer were treated with immune checkpoint inhibitors alone or in combination. Although the author's experimental design was reasonable and the results were clearly displayed, the study failed to reveal meaningful positive findings and failed to provide clues related to the ineffectiveness of anti-TIM3 treatment, thus the significance is limited.

Comments on the Quality of English Language

English is fine.

Author Response

Reviewer 1:

  • In this article, mice with ovarian cancer were treated with immune checkpoint inhibitors alone or in combination. Although the author's experimental design was reasonable and the results were clearly displayed, the study failed to reveal meaningful positive findings and failed to provide clues related to the ineffectiveness of anti-TIM3 treatment, thus the significance is limited.

We thank the reviewer for their constructive review of our research article. Although we appreciate the reviewers opinion, we believe it is essential to also communicate failing/negative results to the scientific community. This would decrease the unnecessary repeat of certain experiments and subsequent use of animals. Additionally, these results could be important to optimize clinical trial designs testing immune checkpoint blockade in the future.

In reference to the clues related to the ineffectiveness of the anti-TIM3 treatment, we would like to refer to the discussion in which we are referring to other studies testing the effect of anti-TIM3 treatment. The study of Guo et al. showed beneficial results but was tested at earlier disease stages. This suggests a possible treatment strategy in patients diagnosed at earlier stages in contrast to our study in which the therapy was tested in advanced stages of ovarian cancer (which is unfortunately still the clinical reality). Secondly, a potential explanation for the lack of response after anti-TIM3 treatment in our study could be linked to the high immunosuppressive state observed in the ovarian cancer immune microenvironment (as we and others have demonstrated), which may not be lifted after targeting only one checkpoint interaction. Furthermore, literature suggests that one anti-TIM3 antibody may not provide blocking of all ligand interaction sites on the TIM3 receptor which can also partially explain poor outcomes in our study.

We have elaborated this part in our manuscript a bit more:

Line 439-441: Blocking only one checkpoint receptor may therefore be insufficient to overcome this high level of immunosuppression seen in the ovarian cancer tumor microenvironment.

Because we believe negative results should be communicated to the scientific community and because we have provided clues related to the possible reasons for this outcome, we feel that this article is relevant and significant.

Reviewer 2 Report

Comments and Suggestions for Authors

Ovarian cancer is a one of most deadly cancers in women. There is an unmet need for novel and targeted therapies. Therefore, research in this field is highly essential. In this article authors tested anti-TIM3 therapy along with chemotherapies in murine ovarian cancer model and claim that anti-TIM3 therapy fails to prolong survival of these mice. I have major concerns with experiments in this article which are listed as below:

1: Authors mentioned that from literature review TIM3 expression is slightly lower in ovarian tissues when compared to PD-1. Did authors check if this is true in their cell line or mouse model? If there is no expression what is point of inhibition? Most cells and cell lines transform with time, it would be best to check cell lines at regular intervals and confirm if TIM3 expression is high or low.

2: Authors are inoculating 5 million cells. May be this is too much of cells that at day 20 cancer is already untreatable. Authors also did not provide tumor growth curves, mouse body weight curves. How did authors monitor tumor growth? Bioluminescence? Where are those curves?

Unfortunately, just looking at survival curves its hard to accept conclusions made by authors.

3: Authors showed that with anti-TIM3 therapy mice survival days decreased. Did authors check drug toxicity? What is drug delivery formulation for all these drugs?

4: It would be better if authors check anti-TIM3 drug toxicity first in non-tumor bearing mice.

5: Authors demonstrated with survival curves that there is no difference between Chemo and Chemo+anti-TIM3 treatments. Again, did authors check drug delivery formulations if it is reaching cancer cells? If so, where is Immunostaining of TIM3 expression downregulating in cancer tissues data?

6: Lastly, chemotherapy suppresses immune system by decreasing CD4+, CD8+ and naïve B cells percentage. I would be best to also check TIM3 expression modulation with chemotherapy.

Unfortunately, it is hard to accept the findings made by authors with poor choice of data presentation and experimental methods.

Round 2

Reviewer 1 Report

Comments and Suggestions for Authors

Negative results in cancer treatment experiment can be caused by many factors. The authors failed to address my concerns.

Comments on the Quality of English Language

English is good.

Author Response

In order to refrain from over-interpreting the results in our study, we have further adapted some parts in the discussion and conclusion of our manuscript. We believe our results may indicate that, similarly to anti-PD1 treatment in ovarian cancer, anti-TIM3 therapy may not provide benefit to the entire HGSOC patient population. However, some subpopulation, which are yet to be identified, may benefit from ICI treatment after sufficient additional research. Therefore, we don’t want our results to completely dismiss anti-TIM3 as a potential treatment but rather spread caution that more research and optimization is required before clinical trials are feasible.

Adaptations made in discussion and conclusion:

  • Line 424-426: Overall, the current treatment combination and administration schedules tested in this study did not show a potential of anti-TIM3 therapy for ovarian cancer. This was confirmed by our immunological analysis where only very minor differences could be depicted.
  • Line 473-480: As in all preclinical research, it is imperative to acknowledge the translational gap when using mouse models to study therapeutic response. Moreover, studies have indicated to a lower immune infiltration being present in ID8 bearing mouse compared to other ovarian cancer mouse models [45]. In addition, the ID8-fLuc derived mouse model lacks p53 and BRCA1/2 mutations which have also shown to produce distinct properties and impact the TME. However, we have demonstrated a similar abundant immunosuppression in the ID8-fLuc model as in patients [22], as well as a similar response  to first line chemotherapy and anti-PD1 checkpoint inhibition as seen in clinical trial results so far [48]. Nevertheless, it may be relevant to perform additional testing in models with phenotypic differences to the ID8-fLuc model to simulate the heterogeneity in human cancers and the model independency of preclinical experimental findings.
  • Line 490-498: In conclusion, our results indicate that anti-TIM3 checkpoint inhibition may be insufficient to induce a beneficial outcome as monotherapy for ovarian cancer. Furthermore, the combination schedules with either anti-PD1 or standard-of-care chemotherapy tested in the ID8-fLuc mouse model does not provide the envisaged response. Changing the order of immune-oncological combination treatment can impact the outcome but did not solve the lack of anti-TIM3 efficacy for ovarian cancer treatment in this study. Additional preclinical experiments towards more promising combination regimens could reveal the potential of this type of immunotherapy as a clinical treatment for ovarian cancer patients.

Reviewer 2 Report

Comments and Suggestions for Authors

Thank you for answering my major concerns. In future I would suggest instead of using mice body weight for tumor and ascites monitoring, it would be best to utilize BLI. 

If not BLI did authors measure no. of  ascites and size of ascites observed upon euthanasia? It would be best to include a chart for that.

Author Response

We thank the reviewer for the comments and suggestions.

We have added additional Kaplan-Meier curves in the supplementary material (Supplementary Figure S3, S4 and S5) displaying the survival taking into account the ascites. In these curves, the survival is taken as the point at which ascites is drained for the first time in mice when they reach 32g. This point represents the buildup of ascites and another measure for the disease development. We included this way of analysis in the text as follows:

Adapted:

  • Line 27: Our results showed no improvement of survival and symptom development in ovarian cancer-bearing mice after anti-TIM3 treatment when used as monotherapy, nor when combined with anti-PD1 or standard-of-care chemotherapy (carboplatin/paclitaxel)
  • Line 135-140: Relieve of distress due to ascites accumulation, a symptom of advanced ovarian cancer, was performed by drainage of the ascites fluid when mice reach 32 grams of weight. First ascites drainage was used as measure to evaluate symptom development and disease progression. Additionally, all mice were evaluated for overall survival based on the previously published humane endpoint criteria by Baert et al. [33].
  • Line 236-238: We observed no difference in survival and ascites accumulation between the treated groups nor when compared to the control group (Figure 3A) (Supplementary figure S3 A).
  • Line 248-249: This decrease was not significant when analyzing ascites accumulation between both groups (Supplementary figure S3 B).
  • Line: 266-268: Similarly, no significant difference in ascites accumulation was noted between both chemotherapy only and combination treated mice (Supplementary figure S4).
  • Line 302-311: Similar observations could be made when displaying the first ascites drainage at 32 grams as a measure of survival (Supplementary figure S5 A). Here, no significant difference was noted between both combination treated groups and chemotherapy alone. However, the ascites accumulation appeared to be slower in the simultaneous treated mice compared to mice receiving only chemotherapy, although this was not statistically significant. Interestingly, this experiment showed a trend towards a difference between simultaneously treated and sequentially treated mice when analyzing both survival and ascites accumulation which, although not significant, may indicate to the importance of treatment order.
  • Line 424: However, both combination schedules were unable to significantly increase survival or delay ascites accumulation in mice compared to chemotherapy alone.
  • Line 460-463: This decreased outcome compared to chemotherapy only treated mice was less apparent when evaluating ascites accumulation. However, in this analysis the difference between both combination treatment regimens (simultaneous vs sequential) was prominent as well.
  • Supplementary Figure S3, S4 and S5.